# Giant onsite electronic entropy enhances the performance of ceria for water splitting

S. Shahab Naghavi[1], Antoine A. Emery [1], Heine A. Hansen [2], Fei Zhou [3], Vidvuds Ozolins[4,5] & Chris Wolverton[1]

Previous studies have shown that a large solid-state entropy of reduction increases the thermodynamic efficiency of metal oxides, such as ceria, for two-step thermochemical water splitting cycles. In this context, the configurational entropy arising from oxygen off-stoichiometry in the oxide, has been the focus of most previous work. Here we report a different source of entropy, the onsite electronic configurational entropy, arising from coupling between orbital and spin angular momenta in lanthanide $f$ orbitals. We find that onsite electronic configurational entropy is sizable in all lanthanides, and reaches a maximum value of $\approx 4.7$ $k_B$ per oxygen vacancy for $Ce^{4+}/Ce^{3+}$ reduction. This unique and large positive entropy source in ceria explains its excellent performance for high-temperature catalytic redox reactions such as water splitting. Our calculations also show that terbium dioxide has a high electronic entropy and thus could also be a potential candidate for solar thermochemical reactions.

[1] Department of Materials Science and Engineering, Northwestern University, Evanston, IL 60208, USA. [2] Department of Energy Conversion and Storage, Technical University of Denmark, DK-2800 Kgs. Lyngby, Denmark. [3] Lawrence Livermore National Laboratory, Livermore, CA 94550, USA. [4] Department of Applied Physics, Yale University, New Haven, CT 06520, USA. [5] Yale Energy Sciences Institute, West Haven, CT 06516, USA. Correspondence and requests for materials should be addressed to C.W. (email: c-wolverton@northwestern.edu)

Ceria (CeO$_2$) has been among the very first[1] and most widely studied[2] materials for catalytic and energy applications. It is used in three-way exhaust automotive catalysts[2–6], solid-state fuel cells[7–10], two-step thermochemical water splitting cycles (TWSC)[11–14], low-temperature water–gas shift reactions[15], and several other industrial catalytic applications[16–20]. To a large extent, the performance of ceria in these applications depends strongly on its oxygen storage capacity and facile Ce$^{4+}$/Ce$^{3+}$ redox reaction.

Ceria is a highly promising redox active material for TWSC[14], in which a metal oxide is reduced at high temperature, $T_H$, and subsequently re-oxidized by exposure to H$_2$O at a lower (but still elevated) temperature $T_L$:

$$CeO_{2-\delta_L} \xrightarrow{T_H} CeO_{2-\delta_H} + \frac{\delta_H - \delta_L}{2} O_2, \qquad (1)$$

$$CeO_{2-\delta_H} + (\delta_H - \delta_L)H_2O \xrightarrow{T_L} CeO_{2-\delta_L} + (\delta_H - \delta_L)H_2 \qquad (2)$$

$\delta_L$ and $\delta_H$ are, respectively, the low-temperature ($T_L$ typically around 800 °C) and high-temperature ($T_H$ typically around 1600 °C) oxygen non-stoichiometries[14, 21]. Meredig and Wolverton[22] examined the thermodynamics of these two-step reaction cycles and showed that a key thermodynamic quantity for increased efficiency is a large solid-state entropy of reduction $\left(\Delta S_{red}^{solid}\right)$. Stoichiometric oxides typically have $\Delta S_{red}^{solid} < 0$, so an additional source of entropy in the reduced phase is required for a high performance in TWSC. Among all the studied stoichiometric oxide reactions in ref. [22] $2CeO_2 \rightarrow Ce_2O_3 + \frac{1}{2}O_2$ is the only one that shows $\Delta S_{red}^{solid} > 0$.

The entropy of reduction of Eq. (1) in the TWSC process is conventionally defined as[23]:

$$\Delta S_{red} = \Delta S_{conf} + \Delta S_{vib} + \frac{1}{2} S_{O_2} \qquad (3)$$

where $\Delta S_{conf}$ includes both electronic and ionic configurational entropy of ceria. The electronic part associated with electron or hole localization contributes to the total entropy in the same fashion as the ionic configurational entropy due to vacancies. Both configurational contributions can be determined accurately by Monte Carlo simulations based on a cluster expansion[24]. $\Delta S_{vib}$ is the vibrational entropy of reduction and $\Delta S_{O_2}$ is the gas phase entropy. The latter does not depend on the choice of materials, and at temperatures higher than 1000 K, which is relevant for TWSC, is ~ 15 $k_B$ per oxygen atom[23, 25]—hereafter, all the reported entropic contributions are per one oxygen vacancy, unless stated otherwise. By excluding the gas phase entropy, we have the solid-state entropy of reduction $\left(\Delta S_{red}^{solid}\right)$, which is the material dependent contribution.

The first experimental studies on the entropy of reduction of CeO$_2$ by Bevan et al.[26] and successively Panlener et al.[23] showed that this quantity has a logarithmic dependence on the non-stoichiometry, $\delta$, of CeO$_{2-\delta}$. They associated this behavior with the change in the configurational entropy ($\Delta S_{conf}$) with varying degrees of non-stoichiometry, while vibrational entropy was estimated to be negligible. Grieshammer et al.[27] calculated the $\Delta S_{vib}$ of defect formation in ceria at a fixed non-stoichiometry and found it to be ~ 2.5 $k_B$, which is not negligible, but smaller than other entropic contributions. Gopal et al.[28] did comprehensive calculations using Monte Carlo simulations based on the DFT-derived cluster expansion Hamiltonian. They calculated all configurational and vibrational entropic contributions (see Eq. (3)) for different values of $\delta$ at a temperature of 1480 K and found that the actual configurational entropy is much smaller than that of commonly assumed ideal solution model.

For $\delta > 0.12$ their calculated total entropy agrees with that of experiment, however, in the $\delta$ range of 0.01–0.12 they underestimate the experimental entropy, leading to a 4.5 $k_B$[28] gap between their calculations and the experimental measurement[23].

In the present paper, we demonstrate that particularly in the case of lanthanides, a different type of electronic entropy should be considered. This source of electronic entropy, which becomes important for elements with partially filled $f$ shells arises when electrons can be distributed over a large number of multiplet states. This entropy results only from onsite configurational entropy in extremely localized $f$ orbitals, which stems from the possible configurations associated with occupations of the same atomic orbitals, and is hereafter denoted $\Delta S_{elec}^{onsite}$. Our results, show that the $\Delta S_{elec}^{onsite}$ contributes to the high performance of ceria in TWSC and fills the ≈4.7 $k_B$ gap between previous theoretical and experimental entropy values. We calculate the entropy of reduction $(M^{n+}/M^{(n-1)+})$ for several other lanthanide cations (praseodymium, neodymium, europium, and terbium) that are stable in two valence states and for which reliable spectroscopic data are available (see ref. [29]). We calculate the crystal field (CF) parameters of Ce$^{3+}$ in the host fluorite CeO$_2$ structure, where each cerium atom experiences a cubic crystal field from eight oxygen atoms. The results show that the electronic entropy of reduction of Ce$^{4+}$ is the highest among $f$ elements, even surpassing the calculated configurational entropy[28] at large off-stoichiometry values, thereby explaining the unique entropic properties of the reduction of CeO$_2$.

## Results

**$L - S$ coupling and crystal-field splitting**. Electronic configurational entropy $\left(S_{elec}^{onsite}\right)$ arises from thermal excitations among orbital microstates created by Russel–Saunders ($L$–$S$) coupling. In multi-electron atoms, $L$–$S$ coupling between orbital and spin angular momenta ($L$–$S$) results in the creation of microstates, which are further split by CF interactions. As the interaction of the very contracted $f$-orbitals of the lanthanide ions with the CF is small[30], it is reasonable to first treat the electronic energy levels of the lanthanide oxides by considering only free ion energies and then subsequently apply the crystal field. For $f$-orbitals of lanthanides, defining the electronic configuration by just number of valence electrons ($4f^n$) is far less descriptive than the term symbol described by the $L$–$S$ coupling scheme[31]. In this scheme, coupling of orbital and spin angular momentum results in $^{2S+1}L_J$ term symbols in which $2S + 1$ is the spin-multiplicity, $L$ is the total orbital quantum number and $J$ is the total angular momentum quantum number, ranging from $|L + S|$ to $|L - S|$ by steps of one. Indeed, Hund's rules imply that the term with the largest value of multiplicity ($2S + 1$) is the most stable one. If several terms have the same multiplicity, then the term with the largest $L$ is the most stable one. If the $f$ shell is less than half-filled then the state with the lowest $J$ has the lowest energy. On the other hand, if the $f$ shell is more than half-filled then the state with the highest $J$ has the lowest energy. The degeneracy of each $J$-multiplet is $(2J + 1)$ and the total number of microstates ($m$) for a given term symbol $^{2S+1}L$ is $(2S + 1) \times (2L + 1)$.

When a free ion is placed in a crystal, the CF further splits each of the degenerate $J$ states to several subsets and breaks the spherical symmetry of the $f$-shell charge distribution, depending on the local symmetry of the ionic environment. Here, we used a fully ab-initio method, opposing crystal potential (OCP)[32], to calculate the crystal field parameters of Ce$^{3+}$ in the host fluorite CeO$_2$ structure, where each cation is coordinated to eight oxygen atoms. This method is motivated by the above mentioned lowering of the spherical symmetry of the $d/f$ charge distribution, e.g., to cubic symmetry in CeO$_2$, due to crystal field interactions. OCP iteratively computes and subtracts CF interactions in order

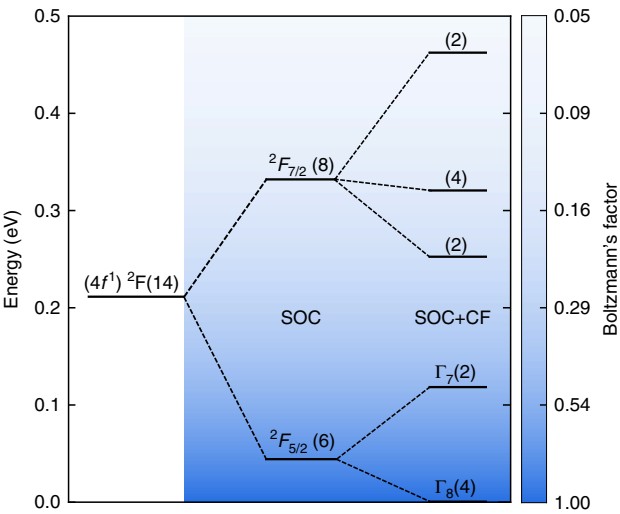

**Fig. 1** Energy levels of the $4f^1$ orbital of $Ce^{3+}$. $Ce^{3+}$ splits initially by spin-orbit coupling (SOC) and subsequently by cubic crystal field (CF) of the the fluorite structure. The spin-orbit splitting between $J = 5/2$ and $J = 7/2$ is about 0.28 eV[29, 49]. The color gradient (see *color bar*) indicates the probability distribution at 1900 K, given by $\exp(-E_i/k_BT)$, and numbers in parentheses stand for the degeneracy of the electronic states. The first predicted $\Gamma_8 \rightarrow \Gamma_7$ excitation for $CeO_2$ is 0.12 eV. Predictions for the higher CF levels of $J = 7/2$ are 0.25, 0.32, 0.46 respectively

to effectively cancel them and recover spherical $d/f$ charge distribution upon convergence. An on-site potential $\lambda_{mm'}$ is introduced as a matrix of Lagrange multipliers in constrained DFT calculations, and crystal field parameters are then obtained as linear combinations of $\lambda_{mm'}$ (see ref. [32] for details). As previously discussed[32], the goal of the constrained DFT calculations was to extract CF parameters rather than to introduce band gap and self-interaction corrections. Consequently the commonly adopted LDA + U approach was not necessary. As a benchmark, the CF parameters of $PrO_2$ from constrained OCP calculations were in good agreement with our previous LDA + U calculations that directly compute the energy difference of different crystal field levels[33].

Figure 1 shows the $f^1$ ($Ce^{3+}$) energy-level splitting scheme in the presence of spin-orbit coupling (SOC) and calculated CF. Without CF, the $f^1$ states split into $^2F_{5/2}$ and $^2F_{7/2}$ separated by approximately 0.28 eV[29]. The CF interaction further splits the six-fold degenerate $^2F_{5/2}$ ground state into a four-fold degenerate $\Gamma_8$ and two-fold degenerate $\Gamma_7$ subsets, separated by 0.12 eV. Crystal field—calculated by OCP method for $Ce^{3+}$—splits the eight-fold degenerate $^2F_{7/2}$ state into states with energies 0.25, 0.32, 0.46 eV. These results are comparable with experimental and theoretical CF splittings of $Ce^{3+}$ doped in $CaF_2$[34–36] and YAG ($Y_3Al_5O_{12}$)[34, 37] hosts.

**Onsite electronic entropy.** By having the energy $E_i$ and degeneracy ($g_i$) of each microstate, $S_{elec}^{onsite}$ of a system with $m$ different microstates can be calculated by

$$S_{elec}^{onsite} = -k_B \sum_i^m g_i p_i \ln p_i \quad (4)$$

where the probability of thermal excitation ($p_i$) to the state with energy $E_i$ is proportional to the Boltzmann factor by

$$p_i = \frac{\exp(-E_i/k_BT)}{Z} \quad (5)$$

| **Table 1 Calculated $S_{elec}^{onsite}$ per oxygen vacancy of selected lanthanide ions before and after reduction at 1900 K** | | | | | |
|---|---|---|---|---|---|
| Element | $f^n$ | Term | Deg. | $S_{elec}^{onsite}$ | $\Delta S_{elec}^{onsite}$ |
| $Ce^{4+}$ | $f^0$ | $^1S$ | 1 | 0.0 | 4.68 |
| $Ce^{3+}$ | $f^1$ | $^2F$ | 14 | 4.68 (4.53)$^{CF}$ | |
| $Pr^{4+}$ | $f^1$ | $^2F$ | 14 | 4.38 (4.22)$^{CF}$ | 1.40 |
| $Pr^{3+}$ | $f^2$ | $^3H$ | 33 | 5.78 | |
| $Nd^{3+}$ | $f^3$ | $^4I$ | 52 | 6.28 | 0.77 |
| $Nd^{2+}$ | $f^4$ | $^5I$ | 65 | 7.05 | |
| $Eu^{3+}$ | $f^6$ | $^7F$ | 49 | 6.59 | −2.43 |
| $Eu^{2+}$ | $f^7$ | $^8S$ | 8 | 4.16 | |
| $Tb^{4+}$ | $f^7$ | $^8S$ | 8 | 4.16 | 2.30 |
| $Tb^{3+}$ | $f^8$ | $^7F$ | 49 | 6.46 | |

Once the $f$-orbitals are occupied, the system gains a large electronic entropy which weakly depends on its occupation number. Therefore, the largest $\Delta S_{elec}^{onsite}$ per oxygen vacancy is associated with the $f^0 \rightarrow f^1$ transition, where fully oxidized state has zero entropic contribution. Entropy units are in $k_B$

and $Z$ is the partition function defined by

$$Z = \sum_i^m g_i \exp(-E_i/k_BT) \quad (6)$$

Equations (4) and (6) indicate that $S_{elec}^{onsite}$ directly depends on two key factors; the probability ($p$) and the total number of microstates ($m$) that varies from 14 in $f^1/f^{13}$ to 66 in $f^5/f^9$ (see Table 1 for the number of degenerate microstates in other occupations). From Ce to Nd, both the total number of microstates and SOC increase. A stronger SOC implies a larger multiplet splitting between the levels which leads to microstates with higher energies. At low temperature, i.e., limited thermal excitations, those high-energy microstates are less probable to be occupied. However, in $Ce^{3+}$ with a moderate SOC, at temperatures relevant for TWSC ($T \approx 1900$ K), a large fraction of microstates are accessible due to the increased thermal excitation, as seen in Fig. 1. Therefore, $S_{elec}^{onsite}$ of $Ce^{3+}$ approaches the ideal limit of $k_B \ln(m)$. As seen in Table 1, $S_{elec}^{onsite}$ weakly depends on the number of electrons and degenerate states. Once the $f$ orbitals get occupied by a single electron—generating 14 microstates with ln (14) $k_B$ ($\approx$2.6 $k_B$) entropic contribution—the system gains a large entropy. For example $S_{elec}^{onsite}$ of $f^1$ ($Ce^{3+}$) is 2.34 $k_B$ and $f^4$ ($Nd^{2+}$) is 3.52 $k_B$.

Myers et al.[38] extracted the electronic entropy contribution of lanthanide ions ($Ln^{3+}$) in lanthanide trihalides from absolute entropy data. Our calculated electronic entropies per ion at $\approx$300 K in units of $k_B$ compared with Myers et al.[38] data (value inside parentheses) are the following: $Ce^{3+}$, 1.79 (1.77); $Pr^{3+}$, 2.19 (2.18), $Nd^{3+}$, 2.30 (2.27); $Eu^{3+}$, 1.13 (1.10); $Tb^{3+}$, 2.56 (2.54). The calculated $S_{elec}^{onsite}$ based on $L$–$S$ coupling shows excellent agreement with previously reported data. Below, we will consider the effect of crystal field splitting on the calculation of $S_{elec}^{onsite}$.

For TWSC applications, the absolute electronic entropy does not matter, only the entropy difference before ($f^n$) and after ($f^{n+1}$) reduction is relevant, $\Delta S_{elec}^{onsite} = 2(S_{elec}^{n-1} - S_{elec}^n)$. The factor two is due to the fact that two $Ce^{4+}$ ions are reduced per oxygen vacancy. As discussed in the previous paragraph and shown in Table 1, because of the large $S_{elec}^{onsite}$ gain upon occupation of an $f$ orbital, in TWSC systems the electronic entropy of reduction reaches its maximum, when the entropy of the oxidized state is zero, $S_{elec}^n = 0$. As a result, the largest $\Delta S_{elec}^{onsite}$ is found in $Ce^{4+} \rightarrow Ce^{3+}$, which undergoes an $f^0 \rightarrow f^1$ redox reaction. Having the oxidized state $f^0$ ($^1S$) with zero onsite electronic entropy is a

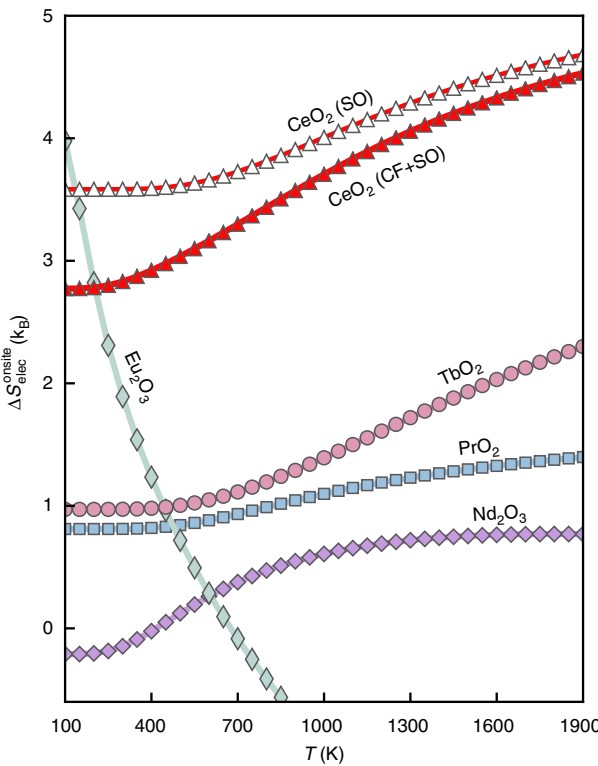

**Fig. 2** Calculated $\Delta S_{elec}^{onsite}$ for lanthanides ions. Predicted electronic entropy of reduction per oxygen vacancy for the lanthanide oxides studied in this work. At high temperature, reduction of $CeO_2$ has the highest $\Delta S_{elec}^{onsite}$ followed by reduction of $TbO_2$ (see Table 1)

**Table 2 Contribution of different entropic terms for $\delta = 0.03$ and temperature of 1500 K**

| Method | $\frac{1}{2}S_{O_2}^0$ | $\Delta S^{vib}$ | $\Delta S^{conf}$ | $\Delta S_{elec}^{onsite}$ | $\Delta S^{tot}$ | $\Delta S_{exp}^{tot}$ |
|---|---|---|---|---|---|---|
| Ideal | 15.2 | 2.5 | 10.4 | – | 28.1 | |
| MC | 15.2 | | 5.9 | – | 21.1 | 26.1 |
| MC + $\Delta S_{elec}^{onsite}$ | 15.2 | | 5.9 | 4.26[a] | 25.4 | |

The values of $\Delta S_{conf}$ are obtained from an ideal solution model and Monte-Carlo (MC) simulations[28]: the MC calculated $S^{conf}$ already includes vibrational entropy[28]. Experimental value is taken from ref. [23]
[a]This value is calculated for $T = 1500$ K

CF only affects the $\Delta S_{elec}^{onsite}$ at low temperatures while at temperatures higher than 1000 K the CF effect is small; therefore, one can just rely on $J$-multiplet states resulting from $L$–$S$ coupling. Table 1 shows that at such elevated temperatures, even pure $S_{elec}^{onsite}$ weakly depends on CF. For example, the effect of CF on $S_{elec}^{onsite}$ of $Pr^{4+}$ and $Ce^{3+}$ is about < 3%. This percentage is in fact an upper limit, as the calculated CF is at DFT lattice parameters at zero Kelvin. The temperature dependent measurement of CF splitting by Walsh et al.[42] showed that the CF splitting significantly decrease with temperature and lattice thermal expansion. Therefore, at 1900 K, which corresponds to typical temperature required for TWSC applications, CF field plays only a minor role.

**Other entropy contributions**. Finally, we compare $\Delta S_{elec}^{onsite}$ with the other sources of entropy. For simplicity we consider a fixed composition of $\delta = 0.03$ roughly corresponding to one oxygen vacancy in a 96-atom supercell. For this composition, we were able to find several reported experimental and theoretical data points (Table 2). At this composition the calculated $\Delta S_{vib}$ is ~ 2.5 $k_B$[27]. The $\Delta S_{conf}$ of $CeO_{2-\delta}$, assuming ideal mixing entropy[23, 28] ($\Delta S_c = -nk_B \ln(\delta)$), where $n$ depends on the defect structure and here $n = 3$, is $\approx 10.4$ $k_B$. However, we note that a system with extensive ordering of oxygen vacancies[43], such as ceria, will have short range order and hence the actual configurational entropy is non-ideal and smaller than the ideal solution model. For instance, the non-ideal $\Delta S_{conf} + \Delta S_{vib}$, calculated by Monte Carlo simulation based on a cluster expansion Hamiltonian, is about 5.9 $k_B$[28], less than half of the ideal $\Delta S_{conf}$. Our calculations show that the neglected electronic entropy ($\Delta S_{elec}$) is > 4.7 $k_B$, which is comparable to these other widely considered sources of entropy and can explain the $\approx 5$ $k_B$ gap between the calculation and experiment. We note that as long as oxygen vacancy is compensated by two polarons $Ce'$ (i.e., $Ce^{4+}$) ($[Ce'_{Ce}] = 2[V_O^{..}]$), $\Delta S_{elec}^{onsite}$ is not a function of off-stoichiometry ($\delta$). Being independent of off-stoichiometry implies that at large $\delta$ the contribution from the electronic entropy surpasses that of the configurational entropy (which decreases with $\delta$) and becomes the major entropy contribution. Using the calculated $\Delta S_{conf}$ in ref. [28], we estimate that this crossover occurs at ($\delta \approx 0.05$).

Our results show that the electronic contribution to the entropy of reduction explains the gap between the results of the currently most detailed theoretical calculations of ref. [28] and the experimental data of Panlener et al.[23] for small deviations from stoichiometry corresponding to $\delta < 0.03$. At larger deviations, adding a constant onsite electronic entropy to the vibrational and configurational entropies from ref. [28] would overestimate the experimental data. There could be several reasons for this apparent discrepancy. For instance, at higher $\delta$ values most of the polarons become bound to oxygen vacancies forming singly charged $V_O^{2-}$–$Ce^{3+}$ or neutral $V_O^{2-}$–$2Ce^{3+}$ complexes[44]; the proximity of $Ce^{3+}$ to an oxygen vacancy could

unique feature of ceria, resulting in a large $\Delta S_{elec}^{onsite}$ of 4.68 $k_B$ per oxygen vacancy, which is a maximum for the reduction of any rare-earth cation. We assert that this unique entropic characteristic of the $Ce^{4+}/Ce^{3+}$ redox reaction helps facilitate the TWSC properties of $CeO_2$. The second largest value of $\Delta S_{elec}^{onsite}$ is found in terbium ($Tb^{4+} \rightarrow Tb^{3+}$) with 2.30 $k_B$ per oxygen vacancy at 1900 K (Fig. 2). There, the non-reduced $Tb^{4+}$ has a half-filled shell with only eight spin-degenerate ($2S + 1$) states but with only one orbital degeneracy ($L = 0$), while $Tb^{3+}$, has an orbital degeneracy of 7 ($L = 3$), providing extra entropy, see Table 1. This extra source of entropy could make $Tb^{4+}$ based materials promising candidates for TWSC applications, as Tb, like Ce, is stable in two valence states ($Tb^{4+}/Tb^{3+}$). This prediction agrees with a recent thermodynamic study that also suggested[39] $TbO_2$ as a potential candidate for TWSC applications.

In contrast to $TbO_2$, reduced $Eu^{2+}$ has a half-filled shell with eight spin- and only one orbital-degeneracy, resulting in a temperature independent entropy of $2 \times \ln(8)k_B = 4.16$ $k_B$ per oxygen vacancy. On the other hand non-reduced $Eu^{3+}$ has a large number of total degeneracy of 49 (see Table 1). Since $\Delta S_{elec}^{onsite} = 2(S_{elec}^{Eu^{2+}} - S_{elec}^{Eu^{3+}})$, by increasing temperature $S_{elec}^{Eu^{3+}}$ rapidly increases which decreases $\Delta S_{elec}^{onsite}$ making it negative at high temperature.

To examine the effect of crystal field on the reduction entropy, we calculated CF parameters for $Ce^{3+}$ in the $CeO_2$ host lattice. We should note that a previous study[40] using inelastic neutron scattering demonstrated that reducing nearest neighbor oxygen atoms have little effect on the overall energy level. This indicates that formation of vacancy next to $Ce^{3+}$ can not dramatically change the CF splitting of a perfect cubic field. Using ab initio CF parameters, previous calculations[41] of $S_{elec}^{onsite}$ for actinides showed an excellent agreement with the available experimental data. Here, we calculate electronic entropy of $Ce^{3+}$. As seen in Fig. 2,

slightly modify the electronic structure and hence reduce the electronic entropy associated with $Ce^{3+}$, but as already discussed the overall effect of oxygen vacancy on the energy levels[40] and electronic entropy is expected to be small. Furthermore, the experimental measurements of Panlener et al.[23] found that the enthalpy of reduction is composition dependent even at very small $\delta$; however, this finding has been challenged due to the large experimental uncertainty[23, 26]. As the entropy is obtained from $T\Delta S = \Delta H - \Delta G$, the entropy values of Panlener et al. may be contaminated by contributions from the composition dependent contribution to $\Delta H$. Indeed, the results of ref. [28] suggest that the entropy stays approximately constant for $\delta$ in the 0.05 to 0.15 range, while the data of Panlener et al.[23] show a pronounced decrease in this range.

Measurements of the Seebeck coefficient provide another means of estimating the electronic entropy contribution in the dilute limit where all polarons are unbound[6, 45]. Unfortunately, the experimental data here are also contradictory. The data of Tuller and Nowick[45] suggest that for small $\delta$ the spin degeneracy factor is one, which contradicts the Kramers theorem requiring that the ground state must be at least doubly degenerate. However, a later study by the same authors[6] concluded that the agreement between the polaron model with spin degeneracy one and the experimental data for the Seebeck coefficient was poor, especially at low $\delta$ where impurities were thought to play an important role. On the theory side, the vibrational entropy of an isolated $Ce^{3+}$ polaron has not been established accurately. Grieshammer et al.[27] have calculated a very large value of about 7 $k_B$ for the entropy of polaron formation at zero pressure, but the largest contribution to this value is due to a volume contribution from the $CeO_2$ host, which was treated in an approximate fashion. Such a large positive entropy is inconsistent with the available data on the Seebeck coefficients in the dilute limit[6, 45]. Hence, thermoelectric measurements on pure, well-equilibrated samples of $CeO_2$ and more accurate calculations of the vibrational entropy associated with free polarons are highly desirable.

## Discussion

We calculated electronic entropies of different lanthanides in the presence of SOC and CF. We calculated CF splittings for $Ce^{3+}$ and $Pr^{4+}$ and found that at temperatures above 1000 K, CF interactions affect the $S_{elec}$ by >3%. The results show that, in ceria, the magnitude of the entropy of reduction due to the commonly neglected onsite electronic entropy ($\Delta S_{elec}$) reaches a maximum of 4.7 $k_B$ per oxygen vacancy, which is twice as large as the vibrational entropy contribution and can be larger than the configurational entropy. This surprisingly large entropy is the result of the very unique electronic structure of cerium in ceria, where redox reactions change its electronic state from $f^0$ to $f^1$. These entropic properties, together with the excellent chemical stability and tolerance for large non-stoichiometry, put ceria in a unique position for two-step solar thermochemical $CO_2/H_2O$ splitting cycles. In addition, we find that Tb (IV) based materials have the next highest electronic entropy, for $Tb^{4+} \rightarrow Tb^{3+}$ redox reactions. We therefore propose compounds containing $Tb^{4+}$ should be experimentally investigated as promising candidates for TWSC applications.

## Methods

Constrained DFT calculations, required for the calculation of crystal field parameters using OCP method, were performed in the VASP package[46] using the Perdew–Becke–Ernzerhof[47] functional, projector augmented-wave potentials[48], a $6 \times 6 \times 6$ k-point mesh an energy cut-off of 520 eV and a convergence criterion of $10^{-10}$ eV/atom for the electronic structure to ensure satisfactory convergence of CF parameters. One electron was added to the neutral $CeO_2$ cell in order to model $Ce^{3+}$. Note that as previously discussed[32], the goal of the constrained DFT

calculations was only to extract CF parameters rather than to introduce band gap and self-interaction corrections.

**Data availability**. The data that support the findings of this study are available within the paper or from the corresponding author on request.

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

## Acknowledgements

S.S.N., A.A.E., C.W., V.O. were supported by US Department of Energy, Office of Science, Basic Energy Sciences, under grant DEFG02-07ER46433. The work of F.Z. was performed under the auspices of the U.S. Department of Energy by Lawrence Livermore National Laboratory under Contract No. DE-AC52-07NA27344.

## Author contributions

All authors contributed to writing and editing the paper.

## Additional information

**Competing interests:** The authors declare no competing financial interests.

