## [Peer Review File · Nature Communications]

Reviewers' comments:

Reviewer #1 (Remarks to the Author):

The paper entitled "Giant Onsite Electronic Entropy Enhances the Performance of Ceria for Water Splitting" calculates the contribution of electronic entropy for the formation of a polaron in lanthanide oxides due to the splitting of f-orbital energy levels. The work is new and interesting from a theoretical point of view, dealing with an effect that has not been addressed in publications so far. Therefore, the results could be interesting for a broad community of researchers in this field. I recommend for publication after minor revisions.

General questions:

Could you please give some details on the computational methods. Which functional and what precision was applied?

Following that, can you elaborate on the accuracy and reliability of the results? The calculated entropy values are still small and the relative error might be significant. Furthermore, it would be interesting to know about the effort it takes to get these results. Is this method generally applicable for materials and could it be used by any researcher?

What would be the effect of e.g. an oxygen vacancy in nearest neighbour position of a Ce³⁺ ion?

Minor remarks:

p1 left "The entropy of reduction of Eq. (2)" should be Eq. (1)

p1 right "otherwise unless stated." Unless stated otherwise

p2 left "where each cerium atom experiences a cubic crystal field from six oxygen atoms." Cerium ions are eight-fold coordinated

p2 right "This method is based on the idea that when an external on-site non-local potential Δ is exactly the opposite of the crystal field potential is applied, the f-shell charge distribution becomes spherical." This sentence is a little hard to read, please consider rephrasing

p3 right "The factor two is due to the fact that two Ce³⁺ ions are reduced per oxygen vacancy." Actually, Ce⁴⁺ ions are reduced and Ce³⁺ ions are formed.

p4 left "Sconf ." blank space

p4 left "(Table)." Which table?

p4 right "Grieshammer et al.[27] have calculated a very large value of 7 kB" Earlier in the paper you mention a value of 2.5 kB from the same paper. I think this is the more realistic one since the formation of polarons is accompanied by the vacancy formation which partly compensates the large entropy value.

p5 left "tolerance for large non-stoichiometry tolerance" too much tolerance

Reviewer #2 (Remarks to the Author):

The authors investigate the source of an entropic gap ($\sim 5 k_B$) between experimental and

theoretical studies of CeO₂, and find that the previously neglected onsite electronic entropy term fits the picture. This is an important finding and of interest especially for the computational materials science community, since the electronic entropy term may be important in a wide range of other materials. The study is rigorous and well-planned, and thus I am in favour of acceptance for publication in Nature Communications, subject to a few clarifications and corrections as detailed below.

Although the authors cite the OCP method (Ref. 31) as their main approach, a detailed methods section/paragraph specifying the DFT functional, k-point mesh, energy cutoff, software package, U value (if DFT+U is used), whether the vacancy defect supercell is charged or neutral, etc., should be added for the benefit of other researchers who are interested in adopting this approach.

The authors should briefly comment on

- the very different behavior of Eu₂O₃ versus the other lanthanide oxides (Fig. 2); and
- the sensitivity of their entropy calculations with respect to lattice constant, since 0 K lattice constants were used for high temperature entropy calculations.

Since the highlight of the paper is really CeO₂, the purpose of including calculations of the other lanthanides may be overshadowed. If there is still room, more discussion on the selection criteria of these other materials and comparison with previous literature would be desirable.

The text in its present form contains many inconsistencies/typos that should be corrected before publication. The following is an incomplete list and I'd like to ask the authors to proofread their manuscript more carefully:

- Use either "on-site" or "onsite" consistently throughout the paper
- Use [element]^[n]+ consistently, instead of [element]^{+[n]}
- Eq. (1): The subscript H in T_H should not be italicized
- Last line on left column, p.1: I believe the entropy of reduction should refer to Eq. (1), not (2)
- 2nd line on left column, p.2: "Eq.3" should be "Eq. (3)"; likewise with "Eqs. 4 and 6"
- Middle paragraph of left column, p.2: Missing "+" after "M^{n-1}"; "Ce^{+4}" should be "Ce^{4+}"
- Last line on right column, p.2: Remove hyphens in "Ce-to-Nd"
- Table I: I'd prefer that the authors report entropy per oxygen vacancy instead of per atom, so that the numbers match with Fig. 2 and the main text
- Right column, p.3: Capitalize "table I"; add "k_B" after 2.6 and 2.4
- Left column, p.4: "Table.I" should be "Table I"
- Under section "Other entropy contributions": "(Table)" should be "(Table II)"
- 2nd last line, right column, p. 4: "7 kB" should be "7 k_B"
- Conclusions: rephrase "tolerance for large non-stoichiometry tolerance"

Response to reviewer comments

We thank the referees for the insightful and constructive comments—they helped significantly to improve the quality and readability of our manuscript. Below, we list all the concerns raised by the referees followed by our response to each of them. According to each referee’s comment, we modified the manuscript. The changes in the revised paper are highlighted in blue.

Reviewer #1

1. Could you please give some details on the computational methods. Which functional and what precision was applied? Following that, can you elaborate on the accuracy and reliability of the results? The calculated entropy values are still small and the relative error might be significant. Furthermore, it would be interesting to know about the effort it takes to get these results. Is this method generally applicable for materials and could it be used by any researcher?

— As seen in the “Onsite Electronic Entropy” section of the paper, the $S_{\text{elec}}^{\text{onsite}}$ is an analytical expression. Therefore, its accuracy depends only on input parameters which are either the spectroscopic data determining the splitting between J -multiplet due to $L-S$ coupling or crystal field (CF) parameters. In the present paper, we study the elements for which reliable experimental spectroscopic data are available (see Ref. [1]). Notice in the case of CF calculations (as seen for Ce^{3+} and Pr^{4+}), experimental data are not required as SOC and CF interaction are calculated by *ab initio* calculations.

To compute the CF parameters, we used a new and fully *ab initio* method, named opposing crystal potential (OCP) [2]. This method is implemented in an in-house code that calculates crystal field interaction through constrained DFT calculations, which were performed in the VASP package using the PBE functional, PAW potentials, a $6 \times 6 \times 6$ k -point mesh, an energy cut-off of 520 eV and a convergence criterion of 10^{-10} eV/atom for the electronic structure to ensure satisfactory convergence of CF parameters. The computational cost of the OCP constrained DFT calculations is approximately that of conventional DFT calculations times a prefactor ($\sim 5-8$), mainly to fully converge the Lagrange multipliers. The required convergence tolerance (about 10^{-10} eV/atom) is more strict than conventional DFT but still quite practical. The OCP method itself is applicable to other materials, irrespective of crystal symmetry. The code is available upon request and will be made open source. Previous work showed that CF splitting calculated by OCP method is accurate and stays within 5% of values fitted to experimental measurements [2] indicating the high accuracy of the fully *ab initio* methods in the calculations of CF parameters. Although we did not find experimental data for CF parameters of ceria, we found that our calculated results are comparable with experimental and theoretical CF splittings of Ce^{3+} doped in CaF_2 [3, 4, 5] and YAG ($\text{Y}_3\text{Al}_5\text{O}_{12}$)[3, 6] hosts, as well as PrO_2 . We note that other methods to calculate

CF splitting can be found in the literature ([7, 8, 9, 10, 11, 12, 13, 3]). Using *ab initio* CF parameters, previous calculation [7] of $S_{\text{elec}}^{\text{onsite}}$ for actinides also showed an excellent agreement with the available experimental data.

In the paper, and after performing CF splitting calculations of ceria using OCP, we showed that CF can be safely neglected in the calculation of onsite electronic entropy at temperatures relevant for TWSC applications, i.e., ≈ 2000 K. The reason behind this is the localization and contraction of *f*-orbitals, which leads to weak CF interaction. We showed that neglecting the CF and using only the atomic spectra data [1] (*J*-multiplet) to calculate the onsite entropy at temperature relevant for water splitting, leads to a maximum error of 3% as seen in Figure 2 and Table I of the manuscript.

To ensure the reliability of the calculated $S_{\text{elec}}^{\text{onsite}}$, we compared our results with previous experiment. As added to the revised manuscript and seen in Box.(6), the calculated electronic entropy per ion at ≈ 300 K in units of k_B compared to Myers *et al.* [14] data (value inside parentheses) are the following: Ce^{3+} , 1.79 (1.77); Pr^{3+} , 2.19 (2.18), Nb^{3+} , 2.30 (2.27); Eu^{3+} , 1.14 (1.10); Tb^{3+} , 2.57 (2.54). The calculated $S_{\text{elec}}^{\text{onsite}}$ based on *L-S* coupling shows excellent agreement with previously reported data.

At the end, this method is applicable to any materials; nevertheless, the calculated $S_{\text{elec}}^{\text{onsite}}$ is significant for *f*-orbitals of lanthanides and actinides.

Proper discussions concerning accuracy of OCP and details of method are added to page 2 of the manuscript

2. What would be the effect of e.g. an oxygen vacancy in nearest neighbour position of a Ce^{3+} ion?

— The main effect of nearest neighbor vacancy is in fact on the crystal field splitting. A. Furrer *et al.* [15] studied the crystal-field interaction of rare-earth compounds using inelastic neutron scattering. They found that in the case of Er^{3+} reducing nearest neighbor oxygen atoms have little effect on the overall energy level sequence. Therefore, as *f*-electrons are pretty localized, and the crystal field due to surrounding oxygen ions is weak, a neighboring vacancy would not change things dramatically, at least not at high temperatures where the solar thermal processes happen.

Discussion of the effect of oxygen vacancy were added on page 4 of the manuscript.

Minor Remarks:

- p1 left “The entropy of reduction of Eq. (2)” should be Eq. (1) ✓
- p1 right “otherwise unless stated.” Unless stated otherwise. ✓
- p2 left “where each cerium atom experiences a cubic crystal field from six oxygen atoms.” Cerium ions are eight-fold coordinated. ✓
- p2 right “This method is based on the idea that when an external on-site non-local potential $\lambda_{mm'}$ exactly the opposite of the crystal field potential is applied, the f -shell charge distribution becomes spherical. This sentence is a little hard to read, please consider rephrasing
- p3 right “The factor two is due to the fact that two Ce^{+3} ions are reduced per oxygen vacancy.” Actually, Ce^{4+} ions are reduced and Ce^{3+} ions are formed. ✓
- p4 left “Sconf.” blank space ✓
- p4 left “(Table).” Which table? ✓
- p5 left “tolerance for large non-stoichiometry tolerance” too much tolerance ✓

3. p4 right “Grieshammer et al. [27] have calculated a very large value of $7 k_B$ ” Earlier in the paper you mention a value of $2.5 k_B$ from the same paper. I think this is the more realistic one since the formation of polarons is accompanied by the vacancy formation which partly compensates the large entropy value.

Grieshammer *et al.* [16] reported value of $7 k_B$ for the entropy of polaron formation ($\Delta_f S_{\text{Ce}'_{\text{Ce}}}$) and the $2.5 k_B$ for vibrational entropy of defect formation ΔS_{vib} which is

$$\Delta S_{\text{vib}} = 2\Delta_f S_{\text{Ce}'_{\text{Ce}}} + \frac{1}{2} S_{\text{O}_2} + S_{\text{V}_\text{O}^{2-}} - S_{\text{bulk}}$$

For the their calculations (see Fig. 5 of Ref. [16]), one can see that the large contribution to the calculated entropy of polaron formation is due to a volume expansion of the CeO_2 host. To make the sentence clear, we explicitly mention the entropy of polaron formation before $7 k_B$ inside the manuscript.

Reviewer #2

4. The authors cite the OCP method (Ref. 31) as their main approach, a detailed methods section/paragraph specifying the DFT functional, k -point mesh, energy cutoff, software package, U value (if DFT+ U is used), whether the vacancy defect supercell is charged or neutral, etc., should be added for the benefit of other researchers who are interested in adopting this approach.

—Thanks the referee for pointing out the missing information. We added a paragraph highlighted in blue explaining the OCP method an its accuracy. The similar discussion is also added under the Box (1) on page 1 of this letter.

5. The very different behavior of Eu_2O_3 versus the other lanthanide oxides (Fig. 2).

The sensitivity of their entropy calculations with respect to lattice constant, since 0 K lattice constants were used for high temperature entropy calculations.

We agree with referee that the very different trend of Eu_2O_3 should be discussed. Thus, we explained the different behavior of Eu_2O_3 in the manuscript on p. 4. In short, reduction of total number of microstates of $f^6 \rightarrow f^7$ transition is the main reason of getting negative slope for the $\Delta S_{\text{elec}}^{\text{onsite}}$ in Eu_2O_3 . Eu_2O_3 shows a behavior that is opposite to that of TbO_2 and thus we discuss it after our discussion on TbO_2 . Here reduced Eu^{2+} has a half-filled shell with only eight spin-degenerate states resulting in a temperature independent entropy equal to $2 \times \ln(8) k_B = 4.16 k_B$ per oxygen vacancy. On the other hand non-reduced Eu^{3+} has a large number of total degeneracy equal to 49 (see Table I of manuscript). Since $\Delta S_{\text{elec}}^{\text{onsite}} = 2(S_{\text{elec}}^{2+} - S_{\text{elec}}^{3+})$, increase of S_{elec}^{3+} due to temperature leads to a decrease in $\Delta S_{\text{elec}}^{\text{onsite}}$ which ends up being negative.

The lattice thermal expansion at high temperature would mainly decrease the CF splitting as discussed by W. Walsh *et al.* [17] who performed a temperature dependent measurement of CF splitting (see the revised manuscript). However, as shown in Table. I for Ce^{3+} and Pr^{4+} the effect of crystal field on the calculated $S_{\text{elec}}^{\text{onsite}}$ at high temperatures is less than 3%. On the other hand, as reported by P. Dorenbos [18], the crystal field of Eu^{3+} is 20% smaller than Ce^{3+} , indicating that $S_{\text{elec}}^{\text{onsite}}$ of Eu^{3+} weakly depends on the lattice thermal expansion, and coordination number (as already discussed in this letter).

6. Since the highlight of the paper is really CeO_2 , the purpose of including calculations of the other lanthanides may be overshadowed. If there is still room, more discussion on the selection criteria of these other materials and comparison with previous literature would be desirable.

We agree with the referee on this point and thus we added the relevant discussion to the manuscript on page 2 and 4. Here, we used additional elements to mainly show the uniqueness of the $f^0 \rightarrow f^1$ transition happening in ceria in comparison to other possible transitions. We used elements that have reliable spectroscopic data [1] for their two different oxidation states ($M^{n+}/M^{(n-1)+}$). The elements with electronic configuration bigger than f^8 are not considered as the total number of degenerate state decreases by increasing the number of electrons leading to negative entropy of reduction.

To check the accuracy of our results we compared our $S_{\text{elec}}^{\text{onsite}}$ calculations with reported data of Myers *et al.* [14], who extracted the electronic entropy contribution of lanthanide ions (Ln^{3+}) in lanthanide trihalides from measured total entropy. Our calculated electronic entropy per ion at ≈ 300 K in the unit of k_B compared to Mayers *et al.* [14] data (value inside parentheses) are the following: Ce^{3+} , 1.79 (1.77); Pr^{3+} , 2.19 (2.18), Nb^{3+} , 2.30 (2.27); Eu^{3+} , 1.13 (1.10); Tb^{3+} , 2.56 (2.54). As seen the calculated $S_{\text{elec}}^{\text{onsite}}$ based on $L-S$ coupling shows excellent agreement with previously reported data.

Minor Remarks:

- Use either “on-site” or “onsite” consistently throughout the paper. ✓
- Use $[element]^{[n]+}$ consistently, instead of $[element]^{+[n]}$. ✓
- Eq. (1): The subscript H in T_H should not be italicized. ✓
- Last line on left column, p.1: I believe the entropy of reduction should refer to Eq. (1), not (2). ✓
- 2nd line on left column, p.2: “Eq. 3” should be “Eq. (3)”; likewise with “Eqs. 4 and 6”. ✓
- Middle paragraph of left column, p.2: Missing “+” after “ M^{n-1} ”; “ Ce^{+4} ” should be “ Ce^{4+} ”. ✓
- Last line on right column, p.2: Remove hyphens in “Ce-to-Nd”. ✓
- Table I: I’d prefer that the authors report entropy per oxygen vacancy instead of per atom, so that the numbers match with Fig. 2 and the main text. ✓
- Right column, p.3: Capitalize “table I”; add “ k_B ” after 2.6 and 2.4. ✓
- Left column, p.4: “Table.I” should be “Table I”.
- Under section “Other entropy contributions”: “(Table)” should be “(Table II)”. ✓
- 2nd last line, right column, p. 4: “7 kB” should be “7 k_B ”. ✓
- Conclusions: rephrase “tolerance for large non-stoichiometry tolerance”. ✓

References

- [1] A. Kramida, Yu. Ralchenko, J. Reader, and NIST ASD Team. NIST Atomic Spectra Database (ver. 5.3), [Online]. Available: <http://physics.nist.gov/asd> [2015, November 17]. National Institute of Standards and Technology, Gaithersburg, MD., 2015.
- [2] Fei Zhou and Daniel Åberg. Crystal-field calculations for transition-metal ions by application of an opposing potential. Phys. Rev. B, 93(8):085123, 2016.
- [3] Liusen Hu, Michael F Reid, Chang-Kui Duan, Shangda Xia, and Min Yin. Extraction of crystal-field parameters for lanthanide ions from quantum-chemical calculations. J. Phys. Condens. Matter, 23(4):045501, 2011.
- [4] L. van Pieterse, M. F. Reid, R. T. Wegh, S. Soverna, and A. Meijerink. $4f^n \rightarrow 4f^{n-1}5d$ transitions of the light lanthanides: Experiment and theory. Phys. Rev. B, 65:045113, 2002.
- [5] Luke J. Venstrom, Nicholas Petkovich, Stephen Rudisill, Andreas Stein, and Jane H. Davidson. The Effects of Morphology on the Oxidation of Ceria by Water and Carbon Dioxide. J. Sol. Energy Eng., 134(1):011005, 2012.
- [6] Jose Gracia, Luis Seijo, Zoila Barandiarán, Daniel Curulla, Hans Niemansverdriet, and Wouter van Gennip. Ab initio calculations on the local structure and the $4f5d$ absorption and emission spectra of Ce^{3+} -doped YAG. J. Lumin., 128(8):1248–1254, 2008.
- [7] Fei Zhou and Vidvuds Ozoliņš. Self-consistent density functional calculations of the crystal field levels in lanthanide and actinide dioxides. Phys. Rev. B, 85(7):075124, 2012.
- [8] Fei Zhou and Vidvuds Ozoliņš. Crystal field and magnetic structure of UO_2 . Phys. Rev. B, 83(8):085106, 2011.
- [9] Harry Ramanantoanina, Werner Umland, Amador García-Fuente, Fanica Cimpoesu, and Claude Daul. Calculation of the $4f14f05d1$ transitions in Ce^{3+} -doped systems by Ligand Field Density Functional Theory. Chem. Phys. Lett., 588:260–266, 2013.
- [10] P. Novák and M. Diviš. Crystal field parameters of praseodymium in oxides. Phys. status solidi, 244(9):3168–3177, 2007.
- [11] P. Novák, V. Nekvasil, and K. Knížek. Crystal field and magnetism with Wannier functions: Orthorhombic rare-earth manganites. J. Magn. Magn. Mater., 358-359:228–232, 2014.

- [12] P. Novák, K. Knížek, and J. Kuneš. Crystal field parameters with Wannier functions: Application to rare-earth aluminates. Phys. Rev. B, 87(20):205139, 2013.
- [13] Michael Dolg. Computational Methods in Lanthanide and Actinide Chemistry. John Wiley & Sons, 2015.
- [14] Clifford E Myers and Dana T Graves. Vaporization thermodynamics of lanthanide trihalides. J. Chem. Eng. Data, 22(4):440–445, 1977.
- [15] A. Furrer, A. Podlesnyak, M. Frontzek, I. Sashin, J. P. Embs, E. Mitberg, and E. Pomjakushina. Crystal-field interaction and oxygen stoichiometry effects in strontium-doped rare-earth cobaltates. Phys. Rev. B, 90:064426, 2014.
- [16] Steffen Grieshammer, Tobias Zacherle, and Manfred Martin. Entropies of defect formation in ceria from first principles. Physical Chemistry Chemical Physics, 15(38):15935, oct 2013.
- [17] Walter M. Walsh, Jean Jeener, and N. Bloembergen. Temperature-Dependent Crystal Field and Hyperfine Interactions. Phys. Rev., 139(4A):A1338–A1350, 1965.
- [18] P Dorenbos. Crystal field splitting of lanthanide $4f^{n-1}5d$ -levels in inorganic compounds. J. Alloys Compd., 341(1-2):156–159, 2002.